# Single Prolonged Stress Decreases the Level of Adult Hippocampal Neurogenesis in C57BL/6, but Not in House Mice

Ekaterina Kurilova, Maria Sidorova and Oksana Tuchina *

Laboratory for Synthetic Biology, Educational and Scientific Cluster "Institute of Medicine and Life Sciences (MEDBIO)", Immanuel Kant Baltic Federal University, 14 A. Nevskogo Str., Kaliningrad 236016, Russia
* Correspondence: oktuchina@gmail.com

**Abstract:** Many people experience traumatic events during their lives, but not all of them develop severe mental pathologies, characterized by high levels of anxiety that persists for more than a month after psychological trauma, such as posttraumatic stress disorder (PTSD). We used a single prolonged stress protocol in order to model PTSD in long-inbred C57BL/6 and wild-derived (house) female mice. The susceptibility of mice to single prolonged stress was assessed by behavior phenotyping in the Open Field and Elevated Plus Maze, the level of neuroinflammation in the hippocampus was estimated by real-time PCR to TNFα, IL-1β, IL-6, IL-10, Iba1 and GFAP, as well as immunohistochemical analysis of microglial morphology and mean fluorescence intensity for GFAP+ cells. The level of neurogenesis was analyzed by real-time PCR to Ki67, Sox2 and DCX as well as immunohistochemistry to Ki67. We showed that long-inbread C57BL/6 mice are more susceptible to a single prolonged stress protocol compared to wild-derived (house) mice. Stressed C57BL/6 mice demonstrated elevated expression levels of proinflammatory cytokines TNFα, IL-1β, and IL-6 in the hippocampus, while in house mice no differences in cytokine expression were detected. Expression levels of Iba1 in the hippocampus did not change significantly after single prolonged stress, however GFAP expression increased substantially in stressed C57BL/6 mice. The number of Iba+ cells in the dentate gyrus also did not change after stress, but the morphology of Iba+ microglia in C57BL/6 animals allowed us to suggest that it was activated; house mice also had significantly more microglia than C57BL/6 animals. We suppose that decreased microglia levels in the hippocampus of C57BL/6 compared to house mice might be one of the reasons for their sensitivity to a single prolonged stress. Single prolonged stress reduced the number of Ki67+ proliferating cells in the dentate gyrus of the hippocampus but only in C57BL/6 mice, not in house mice, with the majority of cells detected in the dorsal (septal) hippocampus in both. The increase in the expression level of DCX might be a compensatory reaction to stress; however, it does not necessarily mean that these immature neurons will be functionally integrated, and this issue needs to be investigated further.

**Keywords:** PTSD; neurogenesis; house mice; hippocampus; neuroinflammation; cytokines

## 1. Introduction

Scientific studies are usually carried out on laboratory strains of mice and rats for obvious reasons: these animals are genetically homozygous at almost all loci, and this increases the likelihood that scientific results will be reproduced. However, how accurately does the long-inbred mouse strain represent the populations in the wild? Moreover, sometimes a larger genetic diversity is required, for such things as immunological studies [1]. In recent years there has been a growing number of studies comparing genetic, physiological, pharmacological and behavioral characteristics of different inbred and outbred strains as well as wild-derived rodents [2–6]. For instance, the spatial distribution of neural progenitors in the rodent hippocampus depends on mice strain: in C57BL/6 mice, the largest number of proliferating cells and immature neurons are located in the dorsal (septal) hippocampus, while in DBA/2 mice, the number of proliferating cells is approximately

the same in the dorsal and ventral (temporal) hippocampal regions, and the number of immature neurons predominates in the temporal hippocampus [2]. Wiget and coauthors (2017) also showed that in wildly catched house mice, wood mice and bank voles, adult neurogenesis in the hippocampus mostly takes place in the temporal region, while in laboratory strains it is higher in the septal area; among all mice strains studied the highest level of adult hippocampal neurogenesis was observed in C57BL/6 [2]. The level of adult neurogenesis is modulated by various factors, including social interactions, physical activity of an animal, diet and exposure to stress. Stress responses may vary depending on animal strain, sex, and of course the stress protocol itself [7–10]. Single prolonged stress, or the "stress-restress model" was introduced by Liberzon [11] and intensively studied further by other scientists as a rodent model of depression and posttraumatic stress disorder (PTSD) [12–15], especially with respect to female rodents and corresponding sex differences [16]. PTSD is a severe mental pathology which occurs as a delayed or prolonged response to a stressful event of an exceptionally dangerous or catastrophic nature, and is characterized by high levels of anxiety that persists for more than a month after psychological trauma [17,18]. There is strong evidence that PTSD is accompanied by increased levels of pro-inflammatory cytokines such as IL-1β, IL-6 and TNF-α, as well as components of the complement system in the blood (moreover, the levels of the latter are positively correlated with the severity of PTSD symptoms) [19]. PTSD is also known to reduce the level of hippocampal neurogenesis, disrupting the dentate gyrus (DG)–CA3 circuit and pattern separation mechanisms [20,21]. At the same time, low basal levels of adult hippocampal neurogenesis may be a risk factor for PTSD development, since the ablation of neurogenesis affects stress susceptibility [22]. Notably, the effect of fluoxetine, a selective serotonin reuptake inhibitor which is used to treat depression as well as PTSD, seems to target immature neurons specifically in the ventral (temporal) hippocampus [23]. Thus, it is interesting to study how stress affects the neurogenesis level in rodents of different genetic backgrounds, taking into account that spatial distribution of proliferating and immature neurons might be different in laboratory and wild-derived mice. We used the long-inbred mouse strain C57BL/6 and wild-derived (house) female mice in our studies in order to test their susceptibility to a single prolonged stress protocol; the latter was assessed by behavior phenotyping, estimation of cytokine levels, glial markers, reactive gliosis and neurogenesis by real-time PCR and immunohistochemistry.

## 2. Materials and Methods

### 2.1. Animals and Stress Protocol

Adult 2 month old female long-inbred mouse strain C57BL/6 ($n = 12$) and wild-derived (house) mice ($n = 12$) were used in the study. All animals were housed five per cage and maintained in standard environmental conditions ($23 \pm 2\,°C$; 12 h/12 h dark/light cycle) with water and food provided ad libitum in the animal care facility at Immanuel Kant Baltic Federal University in Kaliningrad, Russia. All experiments including the number of animals used in the study were approved by the Independent Ethical Committee of the Clinical Research Center at IKBFU, Kaliningrad, protocol 27/2021. Seven C57BL/6 and six house mice were subjected to single prolonged stress according to the "stress-restress model" of Liberzon [11]. Briefly, animals were fixed in a restrainer for 2 h followed immediately by a forced swim for 10 min in $23 \pm 2\,°C$ water, after which the animals were allowed to rest for 15 min and then were exposed to ether vapors until loss of consciousness. In order to model chronic stress, animals were further fixed in a restrainer for 30 min every 7 days for 1 month, i.e., once a week.

### 2.2. Behavior Phenotyping

Animals were screened using the Open Field (OF) and Elevated Plus Maze (EPM) behavioral tests before and after the stress protocol. Mice that were inactive or showed signs of anxiety during the first screening were removed from the experiment. In order to test the exploratory behavior and general activity in the OF, mice were individually

placed into the corner of an open arena (50 cm × 50 cm × 60 cm height) under dim lighting conditions and allowed to move freely for 5 min. In the OF test, the time spent by the animal in the center and at the periphery of the maze was assessed. We also took into account how many times the mice crossed the center and how much time they spent motionless. In order to analyse anxiety-like behavior in the EPM test, mice were individually placed into the center of a cruciform maze, their head towards an open arm of the maze, and they were allowed to move freely for 5 min. The EPM had two open arms; perpendicular to those arms were two darkened enclosed arms (21 cm length). The four arms meet in a 5-cm square region. In the EPM we assessed how much time the mice spent in the enclosed and open arms of the maze, the number of entries into the center, and how much time they spent motionless. The activity of mice was recorded with a GoPro HERO9 Black camera, and the analysis of behaviour was carried out using BehaviorCloud (Columbus, OH, USA).

### 2.3. Brain Dissection

After experiments, animals were deeply anesthetized with isoflurane (Aesica, Queenborough, UK) and decapitated. Decapitation was carried out after checking the effects of anesthesia (characteristic signs: loss of consciousness, muscle relaxation, lack of pain sensitivity and slow, barely noticeable breathing). Brain isolation procedures were performed in a clean room using medical instruments and a blood disinfectant. After extraction, the brain was cut into hemispheres, and the hippocampus was isolated from one hemisphere for a subsequent polymerase chain reaction. The other hemisphere was fixed in 4% paraformaldehyde in sodium phosphate buffer (PBS, pH 7.4) for at least 24 h and then sectioned on a vibrating microtome (see below).

### 2.4. Real-Time Polymerase Chain Reaction

The hippocampi were isolated from the mouse brain and homogenized using Minilys (Bertin Technologies, Montigny-le-Bretonneux, France). Total RNA was extracted from hippocampal suspension using the ExtractRNA reagent (Evrogen, Moscow, Russia), the amount of RNA was checked under NanoPhotometer Pearl (Implen, München, Germany), and a cDNA synthesis was performed using an MMLV RT Kit (Evrogen, Moscow, Russia) according to supplier protocols. We designed primers using NCBI primer blast software (http://www.ncbi.nlm.nih.gov/tools/primer-blast/, accessed on 4 March 2020) and selected primer pairs with the least probability of amplifying nonspecific products as predicted by NCBI primer blast. Primer sequences were used as follows: IL-1β (FWD-AAAGCTCTCCACCTCAATGG, REV-TGTCGTTGCTTGGTTCTCC), IL-6 (FWD-ATCCAGTTGCCTTCTTGGG, REV-GTCTGTTGGGAGTGGTATCC), TNFα (FWD-TCAGTGTCTTCACCAAAGGG, REV-GCAGTGGACCATCTAACTCG), IL-10 (FWD-GCCCTTTGCTATGGTGTCC, REV-TCTCCCTGGTTTCTCTTCCC), DCX (FWD-CAGCATCTCCACCCAACC, REV-AAGTCCATTCATCCGTGACC), SOX2 (FWD-TGCAGTACAACTCCATGACC, REV-CGGACTTGACCACAGAGC), Aif-1 (Iba1) (FWD-AAGGGAATGAGTGGAAAG, REV-CAGACGCTGGTTGTCTTAGG), GFAP (FWD-ACACTGAAACAGGAGAGAGG, REV-TAAGATGACTGAGCGGATGG), Ki-67 (FWD-AGGACAAGACGTGGGAAAG, REV-TTGCGGGATCGCATAGTT), beta actin (FWD-TGGAATCCTGTGGCATCCATGAAAC, REV-TAAAACGCAGCTCAGTAACAGTCCG). The qPCR was performed with SYBR Green I Real-time PCR dye (ThermoFisher Scientific, Waltham, MA, USA) and a real-time PCR kit (Evrogen, Moscow, Russia) and cycling program of 3 min at 95 °C followed by 45 cycles of 95 °C for 15 s and 63 °C for 35 s (for DCX, Ki67, IL-1B, IL-6, IL-10, TNFα, Beta-actin)/57 °C for 35 s (for SOX2, GFAP, Aif-1 (Iba1), Beta-actin) on a CFX96 Real-Time System (BIO-RAD, Hercules, CA, USA). After completion of qPCR, a melting curve of amplified products was determined. Data were collected using CFX Manager 3.1 (BIO-RAD, Hercules, CA, USA) and analyzed using the ΔΔCt method.

*2.5. Immunohistochemistry*

For immunohistochemistry, one hemisphere was fixed in fresh ice-cold 4% paraformaldehyde solution for at least 24 h, then washed in phosphate buffered saline (PBS), embedded in 5% agarose and then sectioned on a vibrating microtome (Campden Instruments, Loughborough, UK) in order to obtain 50 μm serial sections. Sections were then washed in PBS 3 times for 10 min, and then incubated in 5% fetal bovine serum (FBS) on PBS containing 0.3% Triton X-100 (Sigma-Aldrich, St. Louis, USA, 2725C289) overnight. Sections were divided into strips and incubated with the following primary antibodies: anti-Ki67 rabbit monoclonal (Thermo Fisher Scientific, Waltham, MA, USA, MA514520, 1:1000), anti-Iba1 goat polyclonal (Abcam, Cambridge, UK, ab5076, 1:1000), anti-GFAP rabbit polyclonal (Thermo Fisher Scientific, Waltham, MA, USA, PA1-10019, 1:1000) for 48 h in 1% FBS on 0.3% PBST. After incubation with primary antibodies, the sections were washed three times for 10 min in PBS and incubated with secondary antibodies: donkey anti-rabbit Alexa Fluor 488 (Abcam, Cambridge, UK, ab150073, 1:1000) overnight or horse anti-goat biotinylated (Vector laboratories, Newark, NJ, USA, BA-9500, 1:1000) overnight, following incubation with streptavidin Alexa Fluor 594 conjugate (Thermo Fisher Scientific, Waltham, MA, USA, S11227, 1:500) for 2 h at room temperature. After the staining, all sections were washed three times in PBS for 10 min, mounted on glass slides using a Mowiol-based mounting medium, and examined under an Axio imager A2 fluorescent microscope (Carl Zeiss, Jena, Germany) and LSM 780 confocal scanning microscope (Carl Zeiss, Jena, Germany) with ZEN LSM 780 software. Iba+ cells were counted in the dentate gyrus of the hippocampus, and Ki67+ cells in the subgranular layer of the dentate gyrus. The average mean fluorescence intensity for GFAP was assessed in ZEN software for each brain based on a mean fluorescence intensity of 5 ROI from each hippocampal section.

*2.6. Statistical Analysis*

Data were analyzed using Prism 9 (GraphPad, San Diego, CA, USA). The Shapiro–Wilk test was used in order to determine the normality of the variables, while an unpaired *t*-test was used in order to compare two separate groups with equal variance. The significance threshold was set at $p \leq 0.05$ for all statistical analyses.

## 3. Results

*3.1. C57BL/6 but Not House Mice Show Signs of Increased Anxiety in the Open Field and Elevated Plus Maze*

We assessed the exploratory behavior and general activity of mice in the Open field test (Figure 1A,B). C57BL/6 mice that underwent a single prolonged stress protocol (PTSD group) spent significantly more time immobile than control animals ($p = 0.004$; t = 3.9) and showed signs of anxiety: they spent significantly less time in the center ($p = 0.01$; t = 3.0), more time in the periphery ($p = 0.0003$; t = 5.6) of the maze, and performed less entries to the center ($p = 0.03$; t = 2.5). House mice, however, did not demonstrate any significant differences in behaviour between PTSD and control animals: time immobile ($p = 0.52$; t = 0.6), time in the center ($p = 0.63$; t = 0.4), in the periphery ($p = 0.06$; t = 2.0), entries to the center ($p = 0.56$; t = 0.5) of the maze. However, comparing the data between C57BL/6 and house mice, one can see that house mice behaved similarly to C57BL/6 PTSD animals independent of the stress protocol (C57BL/6 PTSD vs. house control mice: $p = 0.66$, t = 0.4). Mice mobility in the OF was also analysed: the distance covered by stressed C57BL/6 mice was less than corresponding C57BL/6 controls ($p = 0.04$; t = 2.3), but did not differ significantly from house "stressed" ($p = 0.43$; t = 0.8) and house control mice ($p = 0.40$; t = 0.8). There were also no differences between "stressed" and control house mice ($p = 0.88$; t = 0.1). In order to analyze the anxiety-like behavior in more detail, we performed the Elevated Plus Maze test (Figure 1C,D). Stressed C57BL/6 mice have spent significantly more time immobile than control animals ($p = 0.003$; t = 3.9), and they rarely explored open arms and stayed there ($p \leq 0.0001$; t = 7.9) or in the center of the maze ($p = 0.002$; t = 4.2), preferably spending time in the enclosed arms ($p \leq 0.0001$; t = 6.6). Interestingly, house

mice again did not show any significant differences in behavior between "stressed" and control animals: time immobile ($p = 0.98$; t = 0.01), time in the center ($p = 0.27$; t = 1.1), in the open ($p = 0.48$; t = 0.7) and enclosed ($p = 0.13$; t = 1.6) arms of the maze. However, in the Elevated Plus Maze, house mice behaved more like C57BL/6 controls (C57BL/6 control vs. house control mice: $p = 0.79$, t = 0.2). The distance covered in EPM by stressed C57BL/6 mice was less than with the corresponding C57BL/6 controls ($p \leq 0.0001$; t = 9.0), house "stressed" ($p = 0.0009$; t = 4.8) and house control mice ($p = 0.001$; t = 4.7), while there were no differences between "stressed" and control house mice ($p = 0.58$; t = 0.5). Overall, behavior phenotyping of mice in OF and EPM demonstrated that long-inbread C57BL/6 animals show decreased exploratory behavior and signs of anxiety after a single prolonged stress protocol, thus they are more susceptible to this type of stress compared to wild-derived (house) mice.

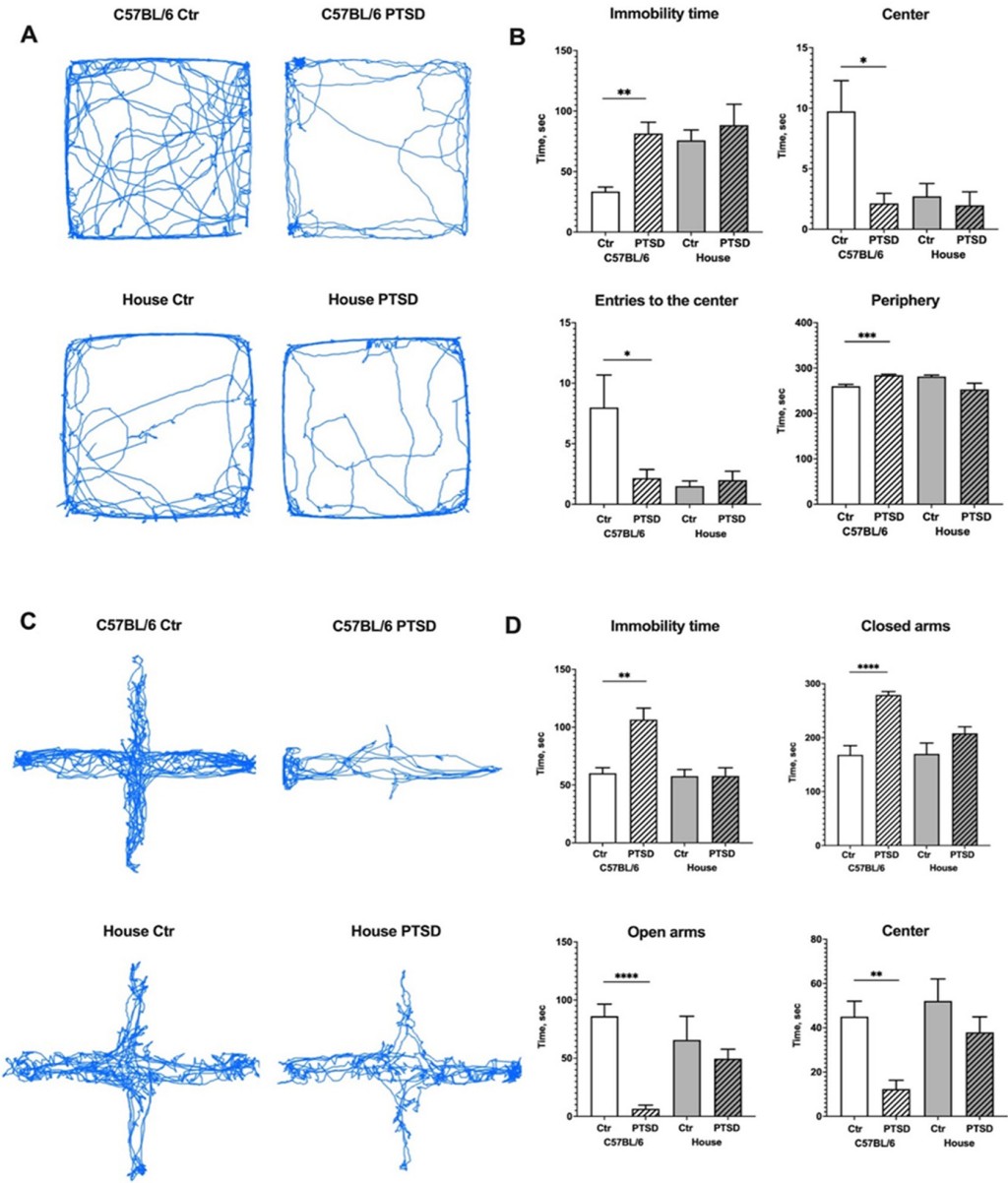

**Figure 1.** Effect of single prolonged stress protocol on the activity of C57BL/6 and house mice in the Open field (**A,B**) and Elevated Plus Maze (**C,D**) tests. (**A,C**)—travel pathway (blue) of illustrative examples of control and stressed animals. PTSD—posttraumatic stress disorder, stressed animals, Ctr—control animals. Data represented as means ± SEM; * $p \leq 0.05$, ** $p \leq 0.01$, *** $p \leq 0.001$, **** $p \leq 0.0001$ (*t*-test).

### 3.2. Cytokine and Glial Markers Profiles Show Signs of Neuroinflammation in the Hippocampi of Stressed C57BL/6 but Not House Mice

A PCR analysis demonstrated that the expression of proinflammatory cytokines significantly increased in the hippocampi of stressed C57BL/6 mice compared to control animals (Figure 2): TNFα ($p \leq 0.0001$; t = 494), IL-1β ($p \leq 0.0001$; t = 47), IL-6 ($p \leq 0.0001$; t = 1241), while the expression of anti-inflammatory IL-10 did not change ($p = 0.59$; t = 0.5). In house mice, however, we did not observe any significant changes in the expression levels between control and "stressed" animals: TNFα ($p = 0.44$; t = 0.7), IL-1β ($p = 0.32$; t = 1.0), IL-6 ($p = 0.78$; t = 0.2), IL-10 ($p = 0.93$; t = 0.08). The expression of microglial marker Iba1 did not change significantly in the hippocampi of C57BL/6 stressed animals compared to control ($p = 0.79$; t = 0.2), which was the same in house mice ($p = 0.62$; t = 0.5), while the expression of astrocytic marker GFAP substantially increased in C57BL/6 stressed animals compared to controls ($p \leq 0.0001$; t = 43), but did not change in house mice ($p = 0.12$; t = 1.7). Thus, a PCR analysis demonstrated that stressed C57BL/6 mice have elevated expression levels of proinflammatory cytokines TNFα, IL-1β, and IL-6, with no differences in the expression level of anti-inflammatory IL-10 in the hippocampus, while in house mice no differences in cytokine expression were detected. Elevated levels of proinflammatory cytokines in the brain indicate ongoing neuroinflammation, the latter of which is probably supported by microglial and astroglial cells.

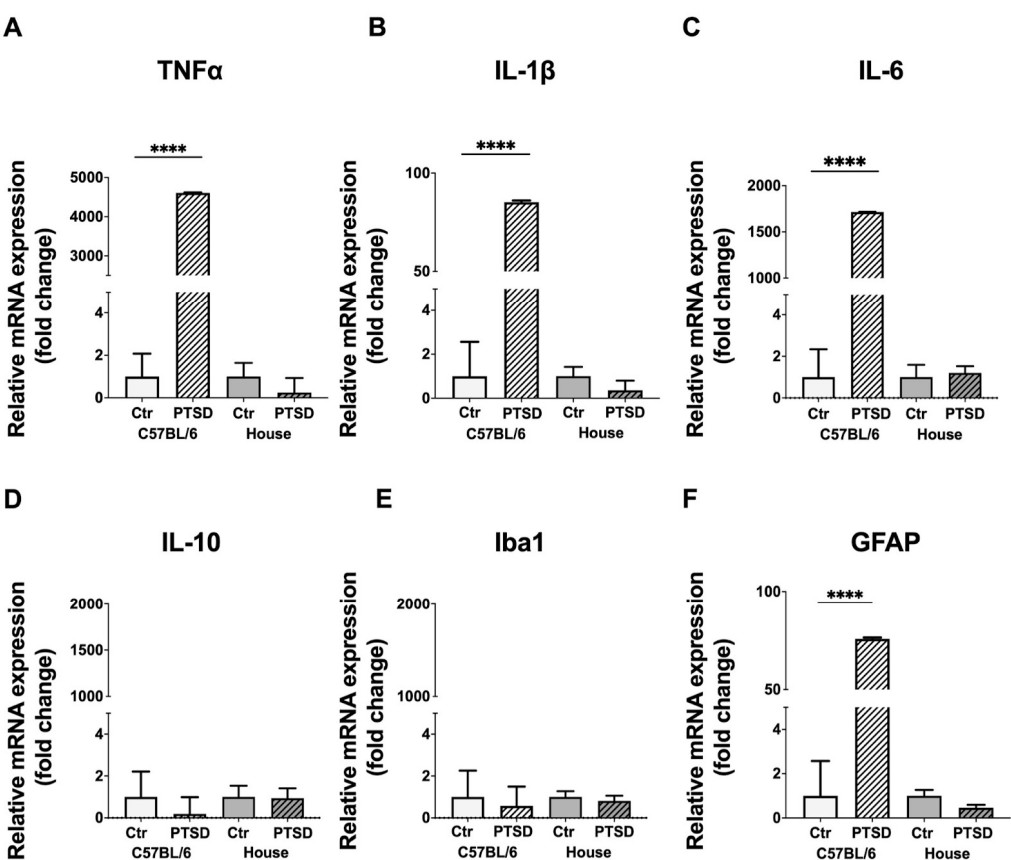

**Figure 2.** Expression levels of pro- and anti-inflammatory cytokines (**A–D**) and glial markers (**E,F**) in the hippocampi of C57BL/6 and house mice in response to single prolonged stress. PTSD—posttraumatic stress disorder, stressed animals, Ctr—control animals. TNFα—tumor necrosis factor α, IL-1β, IL-6, IL-10—interleukins-1β, -6 and -10 correspondingly, Iba1—ionized calcium-binding adapter molecule 1, GFAP—glial fibrillary acidic protein. Data represented as means ± SEM; **** $p \leq 0.0001$ (*t*-test).

The immunohistochemical identification of microglia did not reveal any difference in the number of Iba+ cells in the hippocampi of control and PTSD C57BL/6 ($p = 0.35$;

t = 0.9) and house mice (*p* = 0.71; t = 0.3), however control house mice had significantly more microglia than control C57BL/6 mice (*p* = 0.0002; t = 6.2) (Figure 3A). An analysis of microglial morphology revealed that C57BL/6 mice with PTSD had significantly more rounded amoeboid microglia than C57BL/6 controls (*p* = 0.02; t = 2.7), control house mice (*p* = 0.02; t = 2.7), and "stressed" house mice (*p* = 0.03; t = 2.5). Due to the large number of GFAP+ cells and their complex morphology in the hippocampus, we analyzed the mean fluorescence intensity of selected ROI in the dentate gyrus (Figure 3D). No significant differences were detected between C57BL/6 control versus C57BL/6 PTSD mice (*p* = 0.94; t = 0.07), while "stressed" house mice showed decreased fluorescence intensity compared to control house mice (*p* = 0.003; t = 4.2). Thus, the number of Iba+ cells in the dentate gyrus of C57BL/6 did not change after stress, but their morphology resembled activated ameboid microglia.

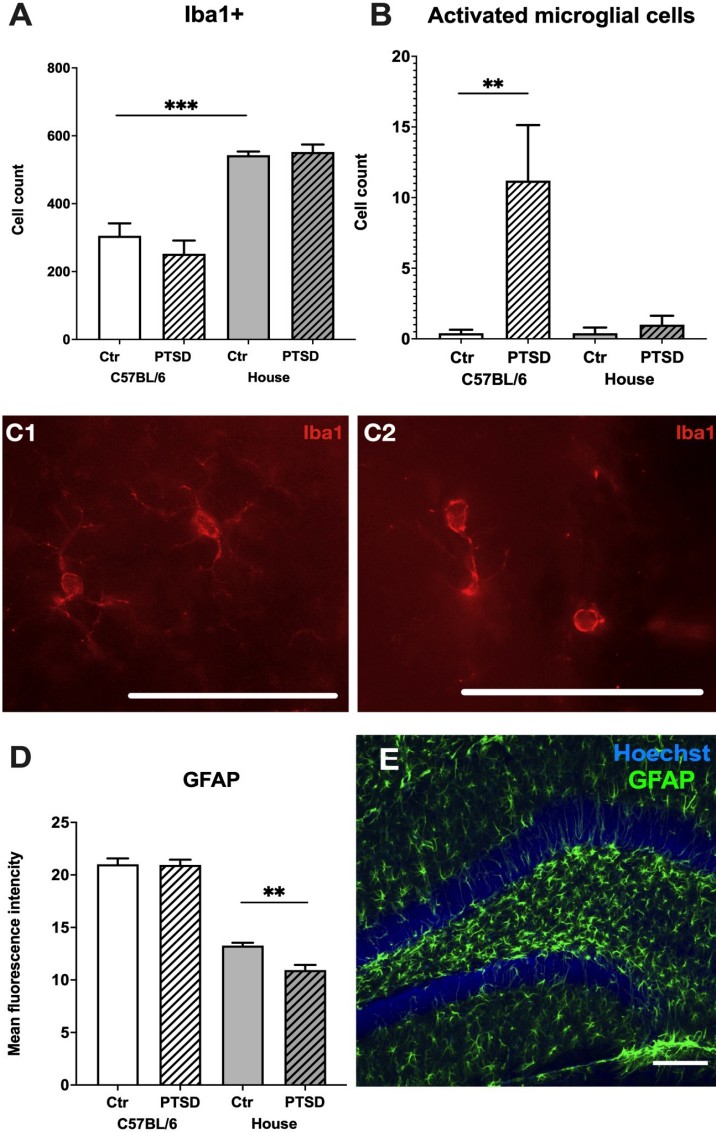

**Figure 3.** Glial profiles in the hippocampi of C57BL/6 and house mice in response to single prolonged stress. PTSD—posttraumatic stress disorder, stressed animals, Ctr—control animals. Number of Iba+ microglial cells (**A**), activated microglia (**B**) and corresponding microphotographs of Iba1+ microglial cells in a resting ramified (**C1**) and activated ameboid (**C2**) states in the dentate gyrus. Scale bar = 200 μm; (**D**)—Mean fluorescence intensity of GFAP+ astrocytes, (**E**)—representative immunostaining of GFAP+ astrocytes in the dentate gyrus. Scale bar = 100 μm. Data represented as means ± SEM; ** *p* ≤ 0.01, *** *p* ≤ 0.001 (*t*-test).

*3.3. Neurogenesis Level Changes Only in C57BL/6 Mice, with the Decrease of Proliferating Cells and Paradoxical Increase in the Expression Level of Immature Neuronal Cell Marker*

Relative mRNA expression levels of Ki67 in the whole hippocampus did not change significantly after stress exposure in C57BL/6 mice ($p = 0.07$; t = 2.0) and in house mice ($p = 0.13$; t = 1.6), which was the same for Sox2 in C57BL/6 mice ($p = 0.42$; t = 0.8) and in house mice ($p = 0.7$; t = 0.3). However, DCX, a marker of immature neurons, showed increased expression in stressed C57BL/6 mice compared to control animals ($p = 0.004$; t = 5.8); while decreased expression in "stressed" house mice ($p = 0.01$; t = 2.9) (Figure 4). Immunohistochemistry with monoclonal antibodies against Ki67 revealed the number of Ki67+ cells in the dentate gyrus to be significantly decreased in stressed C57BL/6 animals in the dorsal hippocampus ($p = 0.01$; t = 3.1), while in house mice no significant changes in cell number were detected ($p = 0.57$; t = 0.5). Ki67+ cells in the subgranular zone of the dentate gyrus demonstrated typical bean shaped morphology and were located in groups (Figure 4E). Thus, stress reduced the number of proliferating cells in the dentate gyrus of the dorsal hippocampus in C57BL/6 mice (but not in house mice), while the expression level of DCX increased significantly in stressed C57BL/6 mice and decreased in "stressed" house mice.

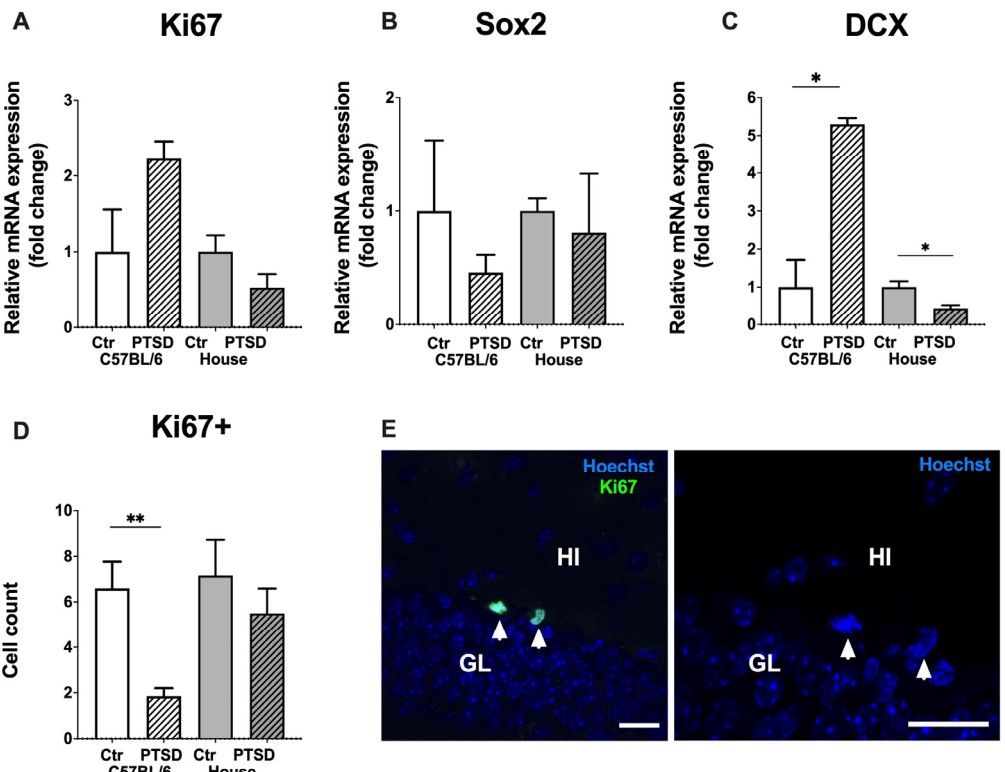

**Figure 4.** Effect of single prolonged stress on the level of neurogenesis in the hippocampus of C57BL/6 and house mice. (**A**–**C**)—relative mRNA expression levels of Ki67, Sox2 and DCX, (**D**)—number of Ki67+ cells in the subgranular layer of the dentate gyrus, (**E**)—representative immunostaining of Ki67+ cells in the dentate gyrus of control C57BL/6 mice, counterstaining with Hoechst. Arrowheads point to Ki67+ newborn cells. GL—granular cell layer, HI—hilus, PTSD—posttraumatic stress disorder, stressed animals, Ctr—control animals. Scale bar = 20 μm. Data represented as means ± SEM; * $p \leq 0.05$, ** $p \leq 0.01$ (*t*-test).

## 4. Discussion

Rodents are naturally aversive to open spaces, and when a mouse is placed into the Open Field arena or EPM it is usually trying to find a shelter. At the same time, the new environment stimulates exploratory behavior, such as locomotion, sniffing, and rears. Thus, defensive behavior competes with the need to gather information about the new

environment. C57BL/6 mice that underwent a single prolonged stress protocol developed anxiety-like symptoms: they spent significantly more time immobile, preferred to stay close to the borders of the Open Field maze/in the enclosed arms of the Elevated Plus Maze, and their exploratory behavior decreased. However, house mice that underwent the same protocol did not show any differences in behavior compared to the corresponding control group, and proved to be more resilient to single prolonged stress. Notably, house mice showed prominent aversion to the center of the Open Field just as C57BL/6 stressed mice did. Since we did not analyze other types of exploratory behavior, such as sniffing and rears, it is highly possible that house mice avoided the center but used other ways to explore new environments, because in the Elevated Plus Maze their behavior resembled the C57BL/6 control group, i.e., there were no signs of anxiety, and the distance covered by "stressed" house mice in OF and EPM did not differ significantly from the corresponding control group. The stress resilience of house mice might be explained by their higher genetic variation compared to the C57BL/6 inbred strain. Genetic risk factors for PTSD are less studied compared to other psychiatric conditions such as bipolar disorder and schizophrenia [24], and are usually related to the dysregulation of the hypothalamic-pituitary-adrenal (HPA) axis or monoamine neurotransmission. One of the interesting candidates is FKBP prolyl isomerase 5 (FKBP5), a co-chaperone of the glucocorticoid receptor (GR) that is highly expressed after stress exposure; it binds to the receptor complex and reduces its sensitivity to cortisol [25]. The overexpression of FKBP5 in C57BL/6 mice induces a stress-like phenotype [26]. Some polymorphisms of FKBP5 have been identified that predispose to stronger mRNA expression, hence HPA axis dysregulation, prolonged cortisol response and modulated activation of brain areas important for fear memory, such as the amygdala [27,28]. Hereditary epigenetic modifications of FKBP5 were observed in Holocaust survivors [29]. Other candidate genes for PTSD development include serotonin, dopamine and cannabinoid receptors (HRT2A, DRD3 and CNR1), dopamine transporter SLC6A3, monoamine transporter SLC18A2 (VMAT2), neuropeptide Y (NPY), C-reactive protein (CRP), retinoid-related orphan receptor (RORA), and the regulator of G-protein signalling (RGS2) [30–35]. Not all studies were conclusive concerning the correlation between certain SNPs and PTSD, especially in humans [35,36]. It is highly likely that the risk of PTSD development is modulated by numerous SNPs, while single polymorphisms have weak effects [35]. Due to technical limitations we were unable to check candidate SNPs in C57BL/6 and house mice in order to understand their different response to stress, but we suppose that the higher genetic variance of house mice protects them from single prolonged stress.

C57BL/6 mice that underwent a single prolonged stress protocol showed elevated expression levels of proinflammatory cytokines TNFα, IL-1β, and IL-6 in the hippocampus, while in house mice no differences in cytokine expression were detected. It is very well known that patients with PTSD usually have chronically elevated levels of TNFα, IL-1β, and IL-6 in their blood [37,38] which is interpreted as a sign of increased inflammation and overall dysregulation of immune response. Other conditions such as major depressive disorder and schizophrenia are also accompanied by increased levels of proinflammatory cytokines in the blood [39,40] as well as in the cerebrospinal fluid [41]. Blood transcriptome studies of patients with PTSD revealed abnormal expression patterns related to processes of innate immune, cytokine response, and type I interferon signaling [42,43]. Elevated levels of proinflammatory cytokines after stress exposure may be due to the hyperactivation of the sympathetic nervous system rather than increased cortisol levels [44–46]. Moreover, sometimes patients with PTSD show decreased cortisol levels (with increased corticotropin-releasing hormone, CRH), i.e., hypoactive HPA axis, and hyperactive sympathetic nervous system [47]. This is in line with the studies showing that increased CRH expression in the brain induces anxiety-like symptoms [48,49]. There were no significant changes in the expression level of anti-inflammatory cytokine IL-10 in C57BL/6 mice, indicating a persistent inflammatory state in stressed animals. It is important to note that we did not perfuse the brains with saline before dissecting the hippocampi, and this may be the

reason why fold changes in the expression levels of TNFα, IL-1β, and IL-6 are quite high in C57BL/6 mice. Thus, cytokines might be expressed by glial cells in the hippocampus as well as by peripheral blood cells. A recent study indicates that hippocampal vascular density differs among several inbred and wild mouse strains, and the sex of analyzed animals is a main factor responsible for transcriptomic differences in the expression patterns of the hippocampal vasculature [50]. Therefore, we can speculate that differences in the permeability of the blood-brain barrier or pericyte-glia signaling might be at least partially responsible for the high expression levels of proinflammatory cytokines in the hippocampus of PTSD C57BL/6 mice.

We have also checked the expression levels of microglial marker Iba1 and astrocytic marker GFAP in the hippocampi of C57BL/6 and house mice. Surprisingly, the expression of Iba1 did not change significantly after a single period of prolonged stress; however, GFAP expression increased substantially in stressed C57BL/6 mice. The number of Iba+ cells in the dentate gyrus also did not change after stress, but house mice had significantly more microglia than C57BL/6 animals. In our previous study with two rat strains with different levels of excitability of the nervous system [51], we have shown that animals of the highly excitable strain had significantly less Iba+ cells in the hippocampus than the less excitable strain [52]. Highly excitable rats seemed to be more susceptible to immune dysfunctions in response to chronic stress. We suppose that the lack of microglia in the hippocampus of C57BL/6 compared to house mice might be one of the reasons for their sensitivity to a single prolonged stress. Despite the fact that there was less microglia in the hippocampus of C57BL/6 mice, stressed C57BL/6 animals were found to have Iba1+ cells in the activated ameboid state, indicating sustained glial activation in response to stress. Chronically activated (reactive) microglia release proinflammatory cytokines and reactive oxygen species which can activate astrocytes [53], the latter of which may induce neuronal death via saturated lipids [54]. The overexpression of GFAP is characteristic of reactive astrocytes [55,56] in Alexander disease [57], Alzheimer's disease [58], major depression [59], and other pathologies. We have identified that the expression level of GFAP in the hippocampus of stressed C57BL/6 mice increased tremendously compared to C57BL/6 controls, and this cannot be explained by the contribution of peripheral blood cells. Thus, hippocampal astrocytes in stressed animals are in an activated state. Analysis of the mean fluorescence intensity of GFAP+ staining in the dentate gyrus did not reveal any differences between stressed and control C57BL/6 mice. It might be possible that astrocytes in other regions of the hippocampus are responsible for high levels of GFAP expression. It must be noted that house mice demonstrated decreased levels of GFAP+ mean fluorescence intensity in the hippocampus compared to C57BL/6 mice. This may be interpreted as both the reduced reactivity of astrocytes in the hippocampus of house mice compared to C57BL/6 animals and the possible decrease of fluorescence intensity due to the drop in argon laser power. However, the decrease in GFAP+ fluorescence intensity in "stressed" house mice compared to the corresponding control cannot be explained by the drop in the laser power since these groups were scanned and analyzed at the same time. Thus, the difference in GFAP+ mean fluorescence intensity between C57BL/6 and house mice must be further estimated.

Chronic stress reduces hippocampal neurogenesis in male mice [59]. In our study, expression levels of Ki67 and Sox2 in the hippocampus did not change after single prolonged stress, either in C57BL/6 or in house female mice. However, the number of Ki67+ cells in the dentate gyrus was significantly reduced in stressed C57BL/6 mice compared to the corresponding control group. Expression levels of Ki67 reflected all proliferating cells in the hippocampus, including the glia and endothelial cells of blood vessels, while immunohistochemistry allowed us to count Ki67+ cells exclusively in the subgranular zone of the dentate gyrus, i.e., only neuronal progenitors. Thus, single prolonged stress reduced hippocampal neurogenesis in female mice, but only in the C57BL/6 strain, not in house mice, and these Ki67+ cells were detected in the dorsal hippocampus. It is known that the spatial distribution of proliferating cells in the hippocampus may depend on the mice/rat

strain. For example, in C57BL/6 mice, the largest number of Ki67+ and DCX+ cells are found in the septal region (dorsal hippocampus), while in DBA/2 mice, the number of Ki67+ cells is approximately the same in the septal and temporal (ventral hippocampus) regions, and the number of DCX+ neurons predominates in the ventral hippocampus [2]. Wiget and coauthors (2017) compared the distribution of proliferating cells and immature neurons in two mice strains (C57BL/6, DBA/2) and the wild-caught species *Mus domesticus*, *Apodemus* sp. and *Myodes* sp.; and in wild animals, the number of Ki67+ cells is either evenly distributed between regions of the hippocampus or predominates in the ventral region, while the largest number of immature DCX+ neurons is observed in the ventral hippocampus. It is known that the dorsal hippocampus is responsible for the formation of spatial memory, while the ventral hippocampus is involved in emotions and motivation [60]. It is likely that these studies testify to the different pressure of natural selection on the processes of neurogenesis in long-inbred mouse strains and wild mice. However, we did not observe any significant changes in the number of proliferating cells in the ventral hippocampus of C57BL/6 and house mice. It is interesting to note that the expression level of DCX increased significantly in stressed C57BL/6 mice. DCX is a marker of immature neurons and its expression level reflects neurogenesis in the adult brain [61]. Since the number of Ki67+ cells in stressed C57BL/6 mice decreased, while the expression of DCX increased it might be possible that single prolonged stress stimulated neuronal differentiation from progenitors in the hippocampus, at least temporarily. In order to prove this hypothesis, the detailed immunohistochemical study with DCX antibodies is needed along with a Western blot analysis of protein levels. It also must be noted that not all differentiating neurons in the end will be incorporated into functional networks of the hippocampus. Thus, we suggest that the increase of DCX expression might be a compensatory reaction to stress; however it does not necessarily mean that these immature neurons will be functionally integrated, and this issue needs to be further investigated.

## 5. Conclusions

We showed that long-inbread C57BL/6 mice are more susceptible to a single prolonged stress protocol compared to wild-derived (house) mice. C57BL/6 but not house mice demonstrate a decrease of exploratory behavior in the Open Field and show signs of increased anxiety in the Elevated Plus Maze.

Stressed C57BL/6 mice demonstrated elevated expression levels of proinflammatory cytokines TNFα, IL-1β, and IL-6 in the hippocampus, while in house mice no differences in cytokine expression were detected.

Expression levels of Iba1 in the hippocampus did not change significantly after single prolonged stress, however GFAP expression increased substantially in stressed C57BL/6 mice. The number of Iba+ cells in the dentate gyrus also did not change after stress, but the morphology of Iba+ microglia in C57BL/6 animals allowed us to suggest that it is activated; furthermore, house mice had significantly more microglia than C57BL/6 animals. We suppose that the lack of microglia in the hippocampus of C57BL/6 compared to house mice might be one of the reasons for their sensitivity to a single prolonged stress.

Single prolonged stress reduced the number of Ki67+ proliferating cells in the dentate gyrus of the hippocampus, but only in C57BL/6 mice, not in house mice, with the majority of cells detected in the dorsal (septal) hippocampus in both. An increase in the expression levels of DCX might be a compensatory reaction to stress; however, it does not necessarily mean that these immature neurons will be functionally integrated, and this issue needs to be investigated further.

**Author Contributions:** E.K. performed all the experiments, M.S. developed SPS protocol and performed supervision, O.T., conceptualization, project administration, funding acquisition, writing—review & editing. All authors have read and agreed to the published version of the manuscript.

**Funding:** This work was supported by the Immanuel Kant Baltic Federal University in Kaliningrad, Russia.

**Institutional Review Board Statement:** All experiments including the number of animals used in the study were approved by the Independent Ethical Committee of the Clinical Research Center at IKBFU, Kaliningrad, protocol 27/2021.

**Informed Consent Statement:** Not applicable.

**Data Availability Statement:** The data presented in this study are available upon request from the corresponding author.

**Conflicts of Interest:** The authors declare that they have no conflict of interest.

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
