# Peer review of "Single Prolonged Stress Decreases the Level of Adult Hippocampal Neurogenesis in C57BL/6, but Not in House Mice"

_cimb, doi:10.3390/cimb45010035_

Round 1
Author Response
Dear Reviewer, thank you for your time and your comments.
- Abstract- The term “lack of microglia” when referring to the C57 mice in comparison to house mice, does not look accurate to me as both types of mice do have Iba1+ cells. I would find more appropriate to say “decreased microglia levels”.
Thank you, you are right and we corrected this in the text.
2. Just as curiosity, how were obtained the wild-derived mouse?
We purchased these mice from a small private company that sells rodents for animal feed in zoos or individuals who keep snakes or birds of prey as pets. We ordered these mice, bred them in our animal facility and performed experiments on their offsprings.
3. Regarding the statistical analysis, if the main goal of the work is to compare C57 vs house-mice, why the authors did not carry out an ANOVA (instead of a t-test) with type (c57 vs house) and stress (singleprolonged stress vs control) as between subjects’ factor and the different behavioural variables as within subjects?
We first checked whether stress protocol induced any changes in C57BL/6 mice, so we did t-test for C57BL/6 control vs C57BL/6 stress and later when we did obtain house mice we did the same for house control vs house stress. Some data we analysed with ANOVA as well which resulted in the same distribution of significance, thus we decided to make multiple t-test comparisons. However if you insist we can re do all statistics using ANOVA.
4. Would be nice to have all the stats values reported in the results section (i.e t values or in the case of the ANOVAs, the F values).
We added all p- and t-values for data represented in Figures in the text.
5. In section 3.3, in the first sentence, I would add “Relative mRNA expression level of Ki67”, so it becomes clearer for the reader that this data refers to PCR assays.
We corrected this sentence.
6. In figure 4 E, is not clear what the images are representing. Are they c57 or house mice or are they control vs PTSD?
Figure 4E shows representative images from the dentate gyrus of control C57BL/6 mice (we specified this in the figure legend). These images are supposed to demonstrate how stained Ki67+ cells (green) in the dentate gyrus looked like in our sections, and that these cells demonstrated typical bean shaped morphology and were located in groups which is consistent with the literature. We suppose that it will be meaningless to show control vs stressed confocal images since in order to make a conclusion one should count all stained cells in all sections (i.e. a couple of images wont represent the statistical conclusion). However we decided to include one scanned image in two separate confocal channels - Ki67+Hoechst and Hoechst alone in order to show that green cells contain nucleic acids (they are stained with Hoechst) and are in fact cells.
Reviewer 2 Report
In the present manuscript, the Authors investigated the capabilities of response to prolongate stress in C57 and wild mice strain.
The authors should better explain the PSTD protocol. I did not understand if the PTSD procedure consisted of 30 minutes a day for a month or four times a month.
Check some inaccuracies on the test (i.e., in 81 lines, "mice" is repeated). I recommend adding the distance covered in OF and EPM tests to understand if the stress does not compromise the mice's mobility. I suggest providing values and p of the tests in the manuscript, which is helpful for the reader.
The paragraphs could be better related to each other, and please try to conclude each chapter with some conclusions.
Author Response
Dear Reviewer, thank you for your time and your comments.
1. The authors should better explain the PSTD protocol. I did not understand if the PTSD procedure consisted of 30 minutes a day for a month or four times a month.
Animals were first subjected to sequential exposure to three stressors (2 h of restraint, a 10 min swim in 23±2°C water, and exposure to ether vapors until loss of consciousness) during a single continuous session. Then in order to model chronic stress, animals were subjected to 30 min restraint every 7 days for 1 month, i.e. once a week or four times a month. We clarified this in the Methods.
2. Check some inaccuracies on the test (i.e., in 81 lines, "mice" is repeated).
Corrected.
3. I recommend adding the distance covered in OF and EPM tests to understand if the stress does not compromise the mice's mobility.
We calculated the data about distances which animals travelled in OF and EPM and represented them in section 3.1 in the results.
4. I suggest providing values and p of the tests in the manuscript, which is helpful for the reader.
We added all p-values (and t-values as suggested by the other reviewer) for data represented in Figures in the text.
5. The paragraphs could be better related to each other, and please try to conclude each chapter with some conclusions.
We added conclusions to each chapter in the results.